# Biocompatibility and Mechanical Properties of Carboxymethyl Chitosan Hydrogels

**DOI:** 10.3390/polym15010144

**Published:** 2022-12-28

**Authors:** Karol K. Kłosiński, Radosław A. Wach, Małgorzata K. Girek-Bąk, Bożena Rokita, Damian Kołat, Żaneta Kałuzińska-Kołat, Barbara Kłosińska, Łukasz Duda, Zbigniew W. Pasieka

**Affiliations:** 1Department of Experimental Surgery, Faculty of Medicine, Medical University of Lodz, Narutowicza 60, 90-136 Lodz, Poland; 2Institute of Applied Radiation Chemistry, Faculty of Chemistry, Lodz University of Technology, Wróblewskiego 15, 93-590 Lodz, Poland; 3Animal House, Faculty of Pharmacy, Medical University of Lodz, Muszyńskiego 1, 90-151 Lodz, Poland

**Keywords:** hydrogel, carboxymethyl chitosan, radiation-induced crosslinking, Live/Dead assay, biocompatibility, in vitro, in vivo, rat model

## Abstract

Hydrogels have the properties of solid substances and are useful for medicine, e.g., in systems for the controlled release of drugs or as wound dressings. They isolate the wound from the external environment and constitute a barrier to microorganisms while still being permeable to oxygen. In the current study, hydrogels were formed from concentrated aqueous solutions of carboxymethyl chitosan (CMCS) via electron beam irradiation, with the presence of a crosslinking agent: poly(ethylene glycol)diacrylate. The aim of the study was to compare the properties and action of biopolymer CMCS hydrogels with commercial ones and to select the best compositions for future research towards wound-dressing applications. The elasticity of the gel depended on the component concentrations and the irradiation dose employed to form the hydrogel. Young’s modulus for the tested hydrogels was higher than for the control material. The Live/Dead test performed on human fibroblasts confirmed that the analyzed hydrogels are not cytotoxic, and for some concentrations, they cause a slight increase in the number of cells compared to the control. The biocompatibility studies carried out on laboratory rats showed no adverse effect of hydrogels on animal tissues, confirming their biocompatibility and suggesting that CMCS hydrogels could be considered as wound-healing dressings in the future. Ionizing radiation was proven to be a suitable tool for CMCS hydrogel synthesis and could be of use in wound-healing therapy, as it may simultaneously sterilize the product.

## 1. Introduction

Hydrogels are materials that have the properties of solid substances—they typically possess a specific shape and have measurable mechanical properties and liquid-like features that allow easy absorption, diffusion, and release of molecules. Hydrogels are valuable for medicine, e.g., in systems for the controlled release of drugs, wound dressings, surfaces, and matrices for tissue culture and engineering [1,2,3,4,5]. Hydrogel dressings are applied to cure burns, pressure sores, ulcers, surgical wounds, and other types of skin lesions. They isolate the wound from the external environment and constitute a barrier for microorganisms while still being permeable to oxygen. They also provide a humid environment conducive to wound healing and prevent the loss of body fluids. Hydrogels have good adhesion to wounds but less so for intact skin, making their removal from the wound painless. These benefits result from three-dimensional structures of mutually linked polymer chains filled with water or water-based solvents [6,7]. To form and sterilize hydrogel dressings, it is possible to use industrially implemented technology involving ionizing radiation. Applying radiation to aqueous solutions of selected polymers allows for the simultaneous crosslinking (hydrogel formation) and sterilization of newly formed products, the latter if a sufficiently high dose is applied. So far, this method is mainly used in manufacturing hydrogel dressings based on biocompatible hydrophilic synthetic polymers [8]. The use of natural polymers as the main component of the hydrogel has many advantages associated with their biocompatibility and biological properties; however, their disadvantages may include imperfect chemical purity, batch-to-batch irreproducibility of the substrate, or the presence of supramolecular structures with limited solubility. The use of hydrophilic substituents (for instance as side moieties or even short side chains) to increase the solubility of certain polysaccharides facilitates the crosslinking of chains by various methods, including ionizing radiation [9,10]. Moreover, the employment of sidechains that are able to polymerize was found as a facile way to form crosslinks between polysaccharide macromolecules, as in an example of dextran methacrylate [11]. Therefore, the longstanding paradigm that radiation may cause only the degradation of polysaccharides is now eradicated [12,13,14].

Chitosan (CS) is an amino polysaccharide that is a deacetylated product of chitin. Chitin is one of the renewable and biodegradable carbohydrate polymers, typically obtained from the exoskeletons of shellfish or insects. CS is composed of N-acetyl-D-glucosamine and D-glucosamine units connected by β-(1-4) bonds [15]. It possesses valuable physicochemical and biological properties and is thus widely used in and investigated for biomedicine, pharmacy, or cosmetic applications—for example, as a wound-healing promoter or drug carrier. The positively charged CS molecule is taken up by the cells, since it interacts with cellular membranes that are negatively charged (about −70 mV), owing to ionic interchanges mediated by the Na^+^/K^+^ pump [16]. The biocompatibility of chitosan carriers is comprehensively summarized in the review by Rodrigues et al. [17]. Due to the biodegradability and low toxicity of chitosan and CS-dependent formulations, they have been used in various in vitro and in vivo experiments that included different cell types (e.g., fibroblasts, keratinocytes, and macrophages) and animal models (e.g., mice, pigs, and rabbits) [18,19,20,21,22]. Amine et al. [23] summarized that CS and its derivatives are useful because they (1) contribute to the in vivo restoration of metallic homeostasis via complexation with, e.g., Cu^2+^, Co^2+^, Ni^2+^, Zn^2+^, and Fe^3+^; (2) contribute to anti-inflammatory and radical scavenger effects; and (3) bind with proteins, extracellular matrix deposits, and mucus through various physical interactions. Both hydrogel and solution forms of CS are currently available, with other forms such as reinforced hydrogels, fibers, and films being of interest when particular mechanical properties are required. CS biomaterials can be optimized using various strategies that include controlling the crystallinity (ratio, orientation, and allomorph type) via the tuning of polymer processing and formulation, or using nanofillers to strengthen the chitosan matrix [24,25,26]. An example study where the mechanical properties of CS hydrogels were thoroughly described was performed by Domengé et al., who concluded that chitosan patches improve the cardiac function in two murine models of cardiomyopathy [27].

CS is soluble only in acidic solutions (pH below circa 5.5), presents an uncontrollable degradation rate, and its hydrogels do not show high water binding capacities [28,29]. In order to circumvent its limited solubility and degradation, many derivatives have been synthesized and investigated, e.g., chitosan esters and N-trimethylene chloride chitosan. One of the chitosan derivatives is carboxymethyl chitosan (CMCS), which has several advantages over its parental chitosan. CMCS possesses an ampholytic character due to amino and carboxyl moieties. Depending on the environment, CMCS may present different ionization states, improving its solubility and widening its use in medicine [30]. Moreover, it exhibits biodegradability, excellent biocompatibility, antioxidant properties, stability in blood, and an ability to form hydrogels while being non-toxic [31]. CMCS hydrogels can be fragile if formed without a crosslinker, such as, i.a., poly(ethylene glycol) diacrylate (PEGDA) or epigallocatechin-3-O-gallate (EGCG) [32,33]. CMCS is used in numerous materials, such as moisture-retention agents, bactericides, wound dressings, artificial tissue, blood anticoagulants, or drug-delivery matrices [34,35]. It is postulated that CMCS can stimulate fibroblasts’ extracellular lysozyme activity, promote normal skin fibroblasts’ proliferation, and inhibit keloid fibroblasts’ proliferation [34]. It is possible to synthesize CMCS hydrogels by radiation initiation upon the processing of highly concentrated aqueous solutions of this carboxymethyl-substituted polysaccharide [36]. Although soft hydrogels, having low gel content and high swelling capacity, are inapplicable for wound dressings, they are ideally suited for soft tissue engineering—for example, as an internal scaffold for a nerve regeneration guide [37]. The structures of CS, CMCS, and PEGDA are visualized in Figure 1.

In this research, PEGDA was used as a crosslinking agent to form CMCS hydrogels by a radiation method. The application of the crosslinker renders hydrogels with tailorable mechanical properties that depend chiefly on the concentration of PEGDA and are very distinct from those weak or even fragile CMCS hydrogels produced without a crosslinker. The reactions between PEGDA and CMCS in order to produce hydrogels were proposed previously by Czechowska-Biskup et al. [38]. Furthermore, these CMCS-based hydrogels showed no cytotoxicity towards mouse fibroblasts, as demonstrated by a 2,3-bis-(2-methoxy-4-nitro-5-sulfophenyl)-2H-tetrazolium-5-carboxanilide (XTT) test [39]. These hydrogels have shown the capability to absorb fluids, which can be controlled by composition, i.e., concentrations of components combined with the applied radiation dose. However, the biological response of human cells and animal organisms to carboxymethyl chitosan hydrogels should be investigated. Thus, the main aim of the research was to confirm the in vitro and in vivo biocompatibility of a new generation of carboxymethyl chitosan hydrogel dressings (produced by a radiation method with the aid of a crosslinker applicable for biomedical purposes), which can be considered as an entry point for future evaluation of the impact on wound healing.

## 2. Materials and Methods

### 2.1. Chemicals/Materials

CMCS of average viscosity 350 kg∙mol^−1^ and weight 440 kg∙mol^−1^ (determined by light scattering), with a deacetylation degree of 93.8% and a substitution degree of 96%, was purchased from Kraeber & Co GmbH (Ellerbek, Germany). High-purity PEGDA (mass weight of 700 g·mol^−1^) and sodium perchlorate monohydrate (NaClO_4_·H_2_O) were purchased from Sigma-Aldrich (Darmstadt, Germany). High-purity water (0.055 µS/cm; TKA MicroPure system) was used in all experiments.

### 2.2. Hydrogel Manufacturing

Three solutions of CMCS, i.e., 3%, 5%, and 10%, were prepared by dissolving the appropriate amount of polymer in a 0.1 mol·dm^−3^ solution of NaClO_4_ and stirring at room temperature for 72 h. Perchlorate’s role as a radiation-resistant salt was to reduce mutually repulsive ionic interactions in CMCS-ionized groups. After the polymer was completely dissolved, 3% or 5% of PEGDA was added. Plastic bags were filled with approximately 1 g of polymer solution and heat-sealed; alternatively, the polymer solution was poured into cylindrical polymer containers with a specific diameter for compression testing. Samples were irradiated at room temperature with a 6 MeV electron beam (EB) from a linear accelerator (Elektronika ELU-6e). As determined by alanine dosimetry, the dose rate was 5 kGy·min^−1^. For hydrogel synthesis, doses of 15, 25, and 35 kGy were used; these were established on the basis of the previous research [39].

### 2.3. Compression Test

Analyses of the mechanical parameters of the hydrogels were conducted in a compression mode using a tensile machine (Z2.5; Zwick Roell, Ulm, Germany) equipped with a 100 N load sensor and computer software for elaborating the results. Experiments were run at room temperature. Hydrogel cylinders of 10 mm diameter and 10 mm height underwent compression testing at the head speed of 1 (mm min^−1^) to the maximum deformation of 60%. Based on the measurements (at least five of them were averaged), the maximum compression stress and compression modulus were determined.

### 2.4. Live/Dead Test

#### 2.4.1. Preparation of Eluates

An appropriate amount of hydrogel was weighed, and a medium for fibroblast culture was added in a proportion of 0.2 g material per 1 mL medium. The sample was incubated at 37 °C for 24 h. When the hydrogels absorbed the medium during the incubation, the eluate level was checked at 2–3 h intervals and supplemented to obtain the final amount of test eluate according to the standard. Finally, the eluates were separated from the hydrogels and filtered through a sterile 0.2 μm-pore filter.

#### 2.4.2. Cell Culture

Human dermal fibroblasts purchased from American Type Culture Collection (ATCC, Manassas, Virginia) were cultured in Fibroblast Basal Medium with a serum-free Fibroblast Growth Kit from the same manufacturer. Cell culture included the usage of phosphate-buffered saline pH 7.4 (PBS; Sigma-Aldrich, Darmstadt, Germany) without calcium and magnesium ions (Thermo Fisher Scientific, Naarden, the Netherlands), antibiotics, i.e., penicillin and streptomycin (Sigma Aldrich, Darmstadt, Germany), trypsin, as well as ethylenediaminetetraacetic acid (EDTA; Sigma-Aldrich, Darmstadt, Germany), and the addition of neutral red.

#### 2.4.3. Cell Viability

The Live/Dead viability/cytotoxicity kit (Invitrogen, Carlsbad, CA, USA) was used in this study [40,41,42,43]. The kit consists of two fluorescent dyes: 4 mM calcein AM in anhydrous dimethyl sulfoxide (DMSO) and 2 mM ethidium homodimer-1 (EthD-1) in DMSO/H_2_O 1:4 (*v/v*). These dyes were chosen because of their ability to specifically bind to the selected cell elements. This kit allows one to distinguish between live and dead cells using two probes that measure cell parameters such as the intracellular activity of esterases and the integrity of plasma cell membranes. Fibroblasts were cultured in 12-well plates (Nunclon™ Delta Surface; Thermo Fisher Scientific, Naarden, the Netherlands). Cells were seeded at 6 × 10^4^ cells/well density in 2 mL of culture medium and incubated for 24 h at 37 °C with 5% CO_2_ and 100% relative humidity. The number of cells was counted using a EVE cell counter (NanoEnTek, Seoul, Korea). After 24 h incubation, the culture medium was exchanged for an experimental medium, i.e., eluates (obtained from hydrogels). The negative control cells contained only the complete medium, while the positive control was obtained after 30 min of cell incubation in the presence of methanol. Cells in the culture medium were incubated for 48 h. The medium was then removed, and cells were washed with PBS buffer twice and stained with markers according to the protocols included in the Live/Dead viability/cytotoxicity kit. Photos were taken using a reverse-phase microscope from the Zeiss Axio Observer D1 equipped with the AxioCam MRc5 digital camera (Zeiss, Jena, Germany). The fluorescence intensity was measured by a Synergy HT microplate reader (Biotek, Winooski, VT, USA) using appropriate excitation and emission filters (for calcein AM: 485/530 nm; for EthD-1: 530/645 nm). The percentages of live cells and dead cells were calculated from fluorescence readings, defined as:$$\%\;Live\;Cells=\frac{F{\left(530\right)}_{sam}-F{\left(530\right)}_{min}}{F{\left(530\right)}_{max}-F{\left(530\right)}_{min}}\times 100\%$$
$$\%\;Dead\;Cells=\frac{F{\left(645\right)}_{sam}-F{\left(645\right)}_{min}}{F{\left(645\right)}_{max}-F{\left(645\right)}_{min}}\times 100\%$$
where: *F*(530)*_sam_*—fluorescence at 530 nm in the experimental cell sample, labeled with calcein AM and EthD-1;

*F*(645)*_sam_*—fluorescence at 645 nm in the experimental cell sample, labeled with calcein AM and EthD-1;

*F*(530)*_min_*—fluorescence at 530 nm in a sample where all (or nearly all = negative control) cells are alive, labeled with EthD-1 only;

*F*(530)*_max_*—fluorescence at 530 nm in a sample where all (or nearly all = negative control) cells are alive, labeled with calcein AM only;

*F*(645)*_min_*—fluorescence at 645 nm in a sample where all the cells are dead, labeled with calcein AM only;

*F*(645)*_max_*—fluorescence at 645 nm in a sample where all the cells are dead, labeled with EthD-1 only.

### 2.5. Animal Studies

#### 2.5.1. Animals

The in vivo studies were conducted on Wistar laboratory rats; males, body weight 300–350 g, age 10–12 weeks. An assessment of the local response to implanted materials was carried out under the requirements of the PN-EN:ISO 10993-6:2016 norm. The species and number of groups were also determined based on the PN-EN:ISO 10993-6:2016 norm (https://www.iso.org/standard/61089.html; accessed on 14 October 2020), according to which the above experiment is designed (the rat is therein indicated as the primary experimental species). The animals were kept in the experimental room, in transparent cages made of polycarbonate, from 3 to 6 pieces per cage, in controlled environmental conditions (with adjustable light cycle 12/12 h, temperature 20–25°C, relative humidity 45–65%, 15–20 air changes per hour). The animals received standard complete livestock feed and drinking water ad libitum. The animals underwent procedures after a 7-day adaptation period. Only healthy and adult animals were used for the studies.

#### 2.5.2. Biocompatibility Test

The specimens were implanted subcutaneously, and the biological response of tissues was assessed in two short-term periods up to 12 weeks (one week in Group I and one month in Group II) and one long-term period exceeding 12 weeks (three months in Group III). Ten healthy animals were assigned randomly to each group. All animals were implanted subcutaneously with 7 specimen types (6 tested materials and 1 control material). The control specimen was a commercially available hydrogel with a structure and physical properties similar to the materials studied, used as a commercial hydrogel wound dressing. For these studies, 6 hydrogel compositions were selected:CMCS 3% and PEGDA 3%;CMCS 3% and PEGDA 5%;CMCS 5% and PEGDA 3%;CMCS 5%and PEGDA 5%;CMCS 10% and PEGDA 2%;CMCS 10% and PEGDA 3%.

The investigated carboxymethyl chitosan and PEGDA concentrations were selected based on our previous research [39]. All samples, prepared in a form compliant with the PN-EN:ISO 10993-6:2016, were placed in subcutaneous pockets on the back of the animal. After the observation period, the samples were removed from the animals and subjected to macroscopic assessment, enabling comparison of tissue response to the tested materials. Each animal was implanted with 7 types of cylindrical samples of diameter and thickness of about 12 mm and 5 mm. Implants were implanted so that they were at least 10 mm away from the edges of the next implant. The indicated number of 30 animals included 3 groups studied according to observation time, each group including 9 animals at a given observation time, i.e., 1 week, 1 month, or 3 months. The remaining 3 out of 30 rats listed in the project were used to supplement the number of test groups given the possible anesthetic difficulties in the surgery and the possible need to refine the method due to the nature of the introduced material, as well as the unexpected mortality of the tested animals. To minimize the number of animals undergoing procedures, individual materials were tested sequentially—first due to the material being tested, and then for observation.

During the procedure, rats were anesthetized with isoflurane. For operational analgesia, 1 mg/kg butorphanol was administered subcutaneously around the neck using a 1 mL syringe with a 0.5 mm needle attached. Before starting surgery, the operator ensured that the animal was anesthetized correctly. A cut along the intervertebral line opened the skin on the animal’s back. The skin was gently cut from the back muscles to a depth of 15 mm on the right and left of the cut line. At equal intervals, a maximum of 4 material samples were placed in separate pockets on the right and left back of the animal so that a total of 10 samples of each material were implanted in the assumed time interval. Each animal had a randomly established implantation pattern (Figure 2). Intradermal pockets were separated by 10 mm interspaces. After inserting the preparation, the pockets were closed with non-absorbent thread to facilitate the location of the examined materials during resection and reduce the influence of the body’s reaction on the breakdown of the absorbed thread. Threads of different colors were used to identify the type of implanted material. The skin and subcutaneous tissue were then joined with one suture to prevent the displacement or folding of the test materials. The edges of the wound were stitched with single, non-absorbable sutures so that the scar was uniform and centered along the spine.

The animals were monitored at least once a day for the appearance of adverse reactions in terms of general health as well as changes at the implant site. To evaluate post-procedural pain, the Rat Grimace Scale [44] was used by veterinarians who were consulted for the outcome. The scale features specific facial action units (e.g., ear or whisker alterations, orbital tightening) that change in response to the procedure. Differences in posture were also evaluated (e.g., crouched posture). After the indicated observation time selected for a given group of animals, euthanasia and implantation site assessment were performed. Euthanasia was performed by intraperitoneal administration using a 22G sodium pentobarbital needle in a volume not exceeding 10 mL/kg body weight. After the death (no corneal reflexes, mydriasis, cessation of breath, and heart rate), the rat was placed on the operating table in a back position so that the dorsal line and subcutaneous implants were clearly visible. Then, the skin on the back was cut along the postoperative scar. Exposed implantation sites were subjected to macroscopic assessment to show possible vascular and tissue changes. Visual lymph nodes (axillary lymph nodes) were also subjected to visual evaluation. The animals were killed using the methods set out in Annex IV to Directive 2010/63/EU of the European Parliament and of the Council of 22 September 2010 on the protection of animals used for scientific purposes. The remains of the animals were utilized according to the procedures in force at the Medical University of Lodz.

### 2.6. Statistical Analysis

Descriptive and statistical inference methods were used to evaluate the data. In the description of measurable variables, arithmetic means, medians, and standard deviations (SD) were calculated. As the distributions of the analyzed variables differed significantly from the normal distribution (as investigated through the Shapiro–Wilk test), the non-parametric ANOVA Kruskal–Wallis test was used. When statistically significant differences were found between the means for successive variables in the compared groups, the post hoc Dunnett pairwise multiple comparison test was used. All tests assumed a statistical significance level of *p* < 0.05.

## 3. Results and Discussion

The results of previous studies indicate that ionizing radiation is an appropriate tool for the synthesis of carboxymethyl chitosan-based hydrogels and their simultaneous sterilization. The tested hydrogels were found to have a significant ability to absorb water (in the 15–200 g range of water retention per 1 g of gel). By selecting the appropriate experimental conditions, such as polysaccharide concentration and cross-linker, hydrogels with the desired properties can be produced [39].

The synthesis of polysaccharide hydrogels with radiation is not straightforward [38]. The process involves transient products from water radiolysis, chiefly hydroxyl radicals, that transfer energy to the polymer by abstracting carbon-bounded hydrogen atoms from the main chain or side groups. Created carbon-centered radicals may transform, initiating various chemical reactions. One involves a single radical, of which the most important are those from the scission of carbon–carbon bonds, or the others, bi-radical in nature, from disproportionation or crosslinking. Radicals located at the main repeating unit of polysaccharides, the anhydroglucose ring, cause degradation through the scission of glycosidic linkages. The radicals located on side moieties, such as carboxymethyl groups, may participate in crosslinking. The decrease in molecular mass results from the former processes, whereas an increase in the molecular mass is observed if the crosslinking reactions prevail. In the latter case, the formation of a network may be eventually achieved. As the polysaccharide hydrogels are typically weak with poor handling properties, multifunctional monomers may be employed to increase crosslinking yield. The terminal double bond of the crosslinker typically reacts with the macroradical of any type. This can be a radical created at the anhydroglucose unit, a radical at the side group, or a radical remaining after the scission of the glycosidic bond. Therefore, the polysaccharide is connected to a linker with the other terminal double bond that is ready for further reaction with the abovementioned macroradicals or another PEGDA macromere. In every case, the molecular weight increases more efficiently than in irradiation without the crosslinker. PEGDA macromere in an initial polymer solution circumventing the radiation-induced degradation of water-soluble cellulose derivatives; therefore, the gels are formed efficiently and may have appropriate tailorable handling properties for various applications.

The basic knowledge derived from the preceding findings allows for the next stages of research, i.e., selecting hydrogels with potentially optimal parameters for animal studies in order to reduce the number and distress of the animals. Nevertheless, the intended use of the produced hydrogels is in the biomedical field, and their safety for the organism is an important factor. Therefore, the mechanical properties and biocompatibility were assessed.

### 3.1. Mechanical Properties Tests

The purpose of conducting mechanical tests was to evaluate the relevant properties and, on their basis, select the best samples for the next stages of the research. The following were calculated: Young’s modulus (Pa); maximum compressive stress (Pa); maximum deformation (mm/mm). Average values were considered for the results of at least five cylindrical samples, each with regard to a combination of component concentrations and radiation dose. Higher concentrations of CMCS were abandoned due to greater cytotoxicity and difficulties in preparing homogeneous mixtures given the high viscosity. Examples of the stress–strain curves and determined parameters are presented in Appendix A.

#### 3.1.1. Young’s Modulus—Flexibility of Hydrogels

Young’s modulus determines the elasticity or stiffness of material in both compression and tension. It expresses the dependence of the relative linear deformation ε of the material on the stress σ that occurs in the range of the elastic deformation. In these tests, the modulus of elasticity was defined as the slope coefficient of the tangent line to the diagram of the stress function to the relative deformation in the initial phase of compression.

The lowest values of Young’s modulus were determined for the lowest concentration of PEGDA (2%) and ranged from 2.3 kPa to 13.3 kPa. The module values at this PEGDA concentration increased in direct proportion to the increase in CMCS concentration and the dose of electron radiation. Several times higher values of Young’s modulus were determined for hydrogels with a PEGDA concentration of 3% than for hydrogels with a concentration of 2%. The modulus values at this concentration increased in direct proportion to the increase in CMCS concentration. On the other hand, the highest values of the modulus were determined for the dose of 25 kGy. The highest values of Young’s modulus were determined for hydrogels with a concentration of 5% PEGDA. The values at this concentration increased in direct proportion to the increase in CMCS concentration. On the other hand, it is inversely proportional to the irradiation dose. The summary is presented in Figure 3.

The Young’s modulus largely depends on the concentration of the components and somewhat on the irradiation dose. This is due to the crosslinking of the material, which is also concentration- and dose-dependent [45]. The control material was a commercial hydrogel dressing available at a pharmacy. Young’s modulus for the control was on average about 12.5 kPa (±3 kPa) and was similar to or slightly higher than that for hydrogels with a 2% PEGDA concentration, lower than for hydrogels with a 3% PEGDA concentration, and significantly lower than for hydrogels with a 5% concentration of PEGDA. This means that hydrogels with a 5% PEGDA concentration and a CMCS concentration above 5%, with a lower radiation dose, have the highest stiffness. The control material was, therefore, less resilient.

#### 3.1.2. Compressive Strength–Maximum Pressure

The maximum compressive strength is the highest value measured, typically appearing just before the destruction of the specimen or at a certain deformation preset by the operator. The value of the maximum pressure for the compression resistance of the CMCS and PEGDA hydrogels depends largely on the concentration of the components and the radiation dose. At a lower concentration of PEGDA, a higher radiation dose increased the strength of the material, while at a higher concentration of PEGDA, greater strength occurred at a lower radiation dose. As with Young’s modulus, the maximum pressure results from crosslinking of the material, which also depends on concentration and dose. The value of the maximum pressure increased with the increasing concentration of the components. The maximum measured pressure for the control material is approximately 280.6 kPa (±35 kPa). The highest values of the maximum pressure were achieved by hydrogels with a concentration of 5% PEGDA and a CMCS concentration above 5% irradiated with a lower radiation dose. It was not the force that disintegrated the control material, but the compression was stopped in order to not damage the device. The tested hydrogel showed values several times lower than the control. The compressive stress test results are summarized in Figure 4.

#### 3.1.3. Deformation at the Max Compression Stress

The value of the maximum displacement of the head in relation to the initial setting was also measured (up to the material disintegration), allowing us to estimate the compressibility (compression) of the tested material. The values given in Figure 5 are the differences between the initial value (the initial height of the material) and the final value (the position of the head at the time of material disintegration).

The PEGDA content had the greatest impact on the compressibility of the material. The hydrogels with the highest PEGDA content showed the greatest stiffness but were destroyed the most quickly under pressure. On the other hand, the gel with the lowest value was definitely more elastic and crumbled; therefore, it was difficult to determine the end of the measurement. CMCS concentration and dose had little effect on hydrogel compression parameters. The values for the individual concentrations and doses are similar, so the trend of changes cannot be determined. The control material was compressed to more than 90% compression and returned nearly to its original state. In general, the gels maintain their dimensions upon compressive stress until they break. The intended application or purpose of the gel is to function as a solid elastic sheet of hydrogel (which may be termed a dressing or plaster, e.g., for wound management applications); therefore, the mechanical performance is mainly of interest. Before conducting clinical trials of the test hydrogels, one may consider adding non-toxic ingredients to increase the compressive strength without deteriorating the other properties [45].

### 3.2. Live/Dead Test

Hydrogels intended for wound treatment, besides proper physical–chemical properties, have to fulfill severe requirements required for medical products used in biological systems. Upon development, product functionality studies in operational environments are typically preceded by biocompatibility testing either in vitro or in vivo [46,47,48,49,50]. Therefore, the CMCS hydrogels were tested to evaluate their cytocompatibility with human fibroblasts. Lower concentrations were used for these studies due to the fact that in our previous research, some cytotoxicity was observed at 15% and 20% concentrations of CMCS [39]. Moreover, handling difficulties in the preparation of these samples due to the high viscosity of the solutions were encountered. For this reason, the investigations of the highest concentrations were excluded from the current approach. Hydrogels resulting from combinations of CMCS and PEGDA in concentrations of 3%, 5%, and 10% of the former, and 2%, 3%, and 5% of the latter were already partially tested in vitro [39].

In order to complete the cytocompatibility examination, the Live/Dead test was carried out in three different concentrations of the extract (i.e., the sol with medium) diluted with the pure medium in proportions of 1:1, 1:10, and 1:100. The eluates were filtered through 0.2 µm pores, and the stock eluate was too viscous, making the filtering difficult as the filter was clogged. According to the aforementioned ISO norm, hydrogel samples release unbound polysaccharide chains into the medium during incubation with the medium. These chains constitute the soluble fraction, i.e., the sol (that, together with the medium, can be considered a gel extract, as the gels are expected to be partially dissolved [39]). The Live/Dead viability/cytotoxicity test was performed to visualize the distribution of living and dead cells after 48 h of incubation with the chosen hydrogel composition (Figure 6). Calcein AM fluoresces green due to the reaction of intracellular esterase and stained live cells, whereas ethidium homodimer binds to the DNA of dead cells and stains them a red color [51]. Images showed that the dilution of the extract increased the number of live cells. The level of dead cells was comparable with cells treated with the diluted extract. For subsequent dilutions, the number of dead cells decreased, and the number of living cells increased. In the 1:1 dilution, the number of dead cells was already negligible at >10%. The results for all dilutions are shown in Table 1 and Appendix A (the amounts of dead cells and live cells do not add up to 100% because they are relative to the control). This confirms the results from previous XTT tests: the CMCS hydrogels are cytocompatible, they do not increase the mortality of cells, and at low concentrations even increase the number of cells in relation to the control, which was a complete culture medium [39]. The compositions with the best biocompatibility were selected for animal testing.

### 3.3. Assessment of In Vivo Biocompatibility

In the course of the experiment, it was intended to find whether the obtained polymers are at least equally as biocompatible as commercial biomaterials and whether it is rational to pursue further research in terms of their clinical applications. Using the ISO standard guidelines, the specimens (exemplary sample in Figure 7A; images obtained by the scanning electron microscopy in Figure 7B) were implanted subcutaneously (Figure 7C), and the biological response of tissues was assessed in two short-term periods up to 12 weeks (one week in Group I and one month in Group II) and one long-term period exceeding 12 weeks (three months in Group III). The control specimen was a commercially available hydrogel with a structure and physical properties similar to the materials studied, used as a commercial hydrogel wound dressing.

In the first days after implantation, rats manifested the features included in the Rat Grimace Scale. Swelling and itching were also common symptoms. After a few days, the undesirable symptoms disappeared, and the post-implantation wounds healed. The rats functioned normally, and their health was assessed as good. One animal died for reasons unrelated to the experiment. After the first week of surgery, the rats began to gain weight normally (Figure 7D). After the observation time, the animals were dissected, which did not show significant undesirable changes in animal organisms and there were no symptoms disqualifying hydrogels for further testing. Macroscopic evaluation only revealed slight hyperemia around some implantation sites (Figure 7E,F). Moreover, the areas of implantation differed with regard to shape (Appendix A). The lack of negative effects of the hydrogels on the animals’ tissues confirms their biocompatibility.

Studies have shown the validity of considering CMCS hydrogels as potential dressings [52,53]. It was shown that gels with a lower content of CMCS and PEGDA were broken down due to the interaction of tissues and body fluids, while gels with a higher content retained their shape. After the first period, most of the hydrogels retained their shape, while in two subsequent periods, only the hydrogels with a PEGDA content of 5% retained their shape, and the others disintegrated. The control hydrogel in each of the tested animals retained its shape and did not disintegrate. Due to the fact that the gels are to be investigated outside the body as wound dressings, a hydrogel with a CMCS concentration of 5% and a PEGDA concentration of 5% should be chosen for further research as, in most cases, it did not break down and retained its shape (as control material).

## 4. Conclusions

The results presented in this report show that ionizing radiation is a suitable tool for the synthesis of carboxymethyl chitosan-based hydrogels. The addition of PEGDA to the CMCS solutions processed by radiation resulted in the formation of hard macroscopic gels, even at relatively low radiation doses of a few kGy, here demonstrated for 15 kGy. In addition, when the applied dose is 25 kGy or more, gels can be formed and sterilized simultaneously in one technological process.

Since the anticipated application of manufactured hydrogels is a biomedical field, safety for the organism is critical. Therefore, as the first step of biocompatibility evaluation, it was decided to assess the viability of the cells that were in contact with gel extracts. The hydrogels showed no cytotoxicity; surprisingly, even an increase in the number of cells was observed relative to the control at the lower concentration of the extract. The biocompatibility studies on the laboratory rat model did not show any inflammatory reactions after a few days, which confirms their biocompatibility with the living organism. Research and analysis showed that gels with a lower content of CMCS and PEGDA were broken down due to the interaction with tissues and body fluids, while gels with a higher content of components retained their dimensions. Therefore, one may use the hydrogel of CMCS crosslinked with PEGDA according to their needs; the consistency of the gel was stable and maintained durability during the entire time of use or decomposed/degraded within a few weeks.

Taking into account the results of this research and the fact that it is planned to assess the relevance of these gels as wound-healing dressings outside the body, a hydrogel with a CMCS concentration of 5% and a PEGDA concentration of 5% should be chosen for further research since, in most cases, it did not break down and retained its shape. The presented results of the mechanical tests of the hydrogels indicate that before starting clinical trials, adding non-toxic ingredients should be considered in order to increase the compressive strength without deteriorating other properties.

## Figures and Tables

**Figure 1 polymers-15-00144-f001:**
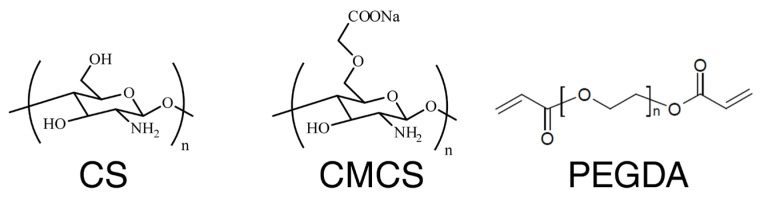
Structure of chitosan (CS), carboxymethyl chitosan (CMCS), and poly(ethylene glycol) diacrylate (PEGDA).

**Figure 2 polymers-15-00144-f002:**
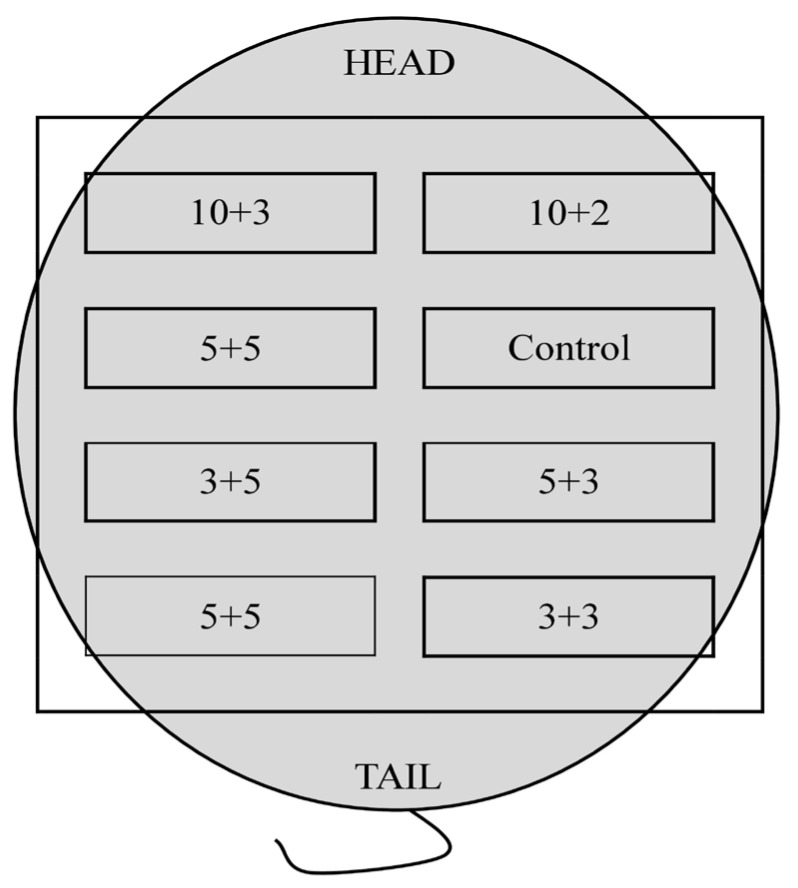
Example scheme of sample implantation. The squares indicate the implantation sites, the numbers indicate the concentration of CMCS + PEGDA, and the “Control” is a commercial hydrogel.

**Figure 3 polymers-15-00144-f003:**
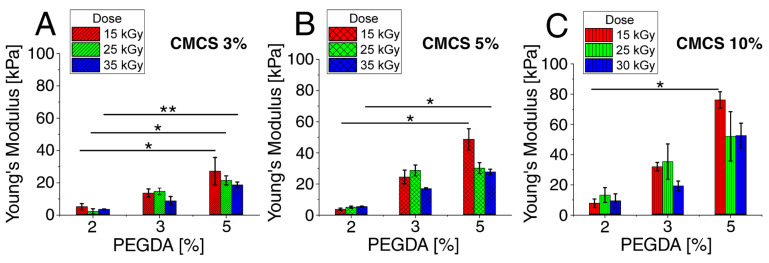
Young’s Modulus (kPa) for hydrogels with various PEGDA and radiation doses for (**A**) CMCS 3%, (**B**) CMCS 5%, and (**C**) CMCS 10%. The *p*-values of the post hoc test are provided; *p* < 0.05 (*), *p* < 0.01 (**). Measurements were made at room temperature (~25 °C).

**Figure 4 polymers-15-00144-f004:**
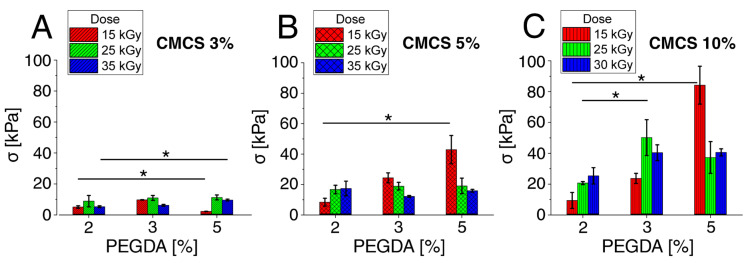
Maximum compressive strength (kPa) for hydrogels with various PEGDA and radiation doses for (**A**) CMCS 3%, (**B**) CMCS 5%, and (**C**) CMCS 10%. The *p*-values of the post hoc test are provided; *p* < 0.05 (*). Measurements were made at room temperature (~25 °C).

**Figure 5 polymers-15-00144-f005:**
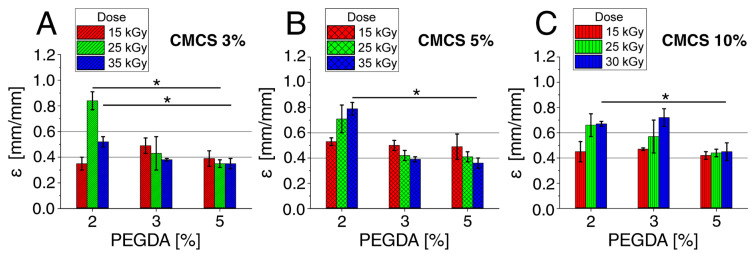
Deformation at maximum compressive strength (mm/mm) for hydrogels with various PEGDA and radiation doses for (**A**) CMCS 3%, (**B**) CMCS 5%, and (**C**) CMCS 10%. The *p*-values of the post hoc test are provided; *p* < 0.05 (*). Measurements were made at room temperature (~25 °C).

**Figure 6 polymers-15-00144-f006:**
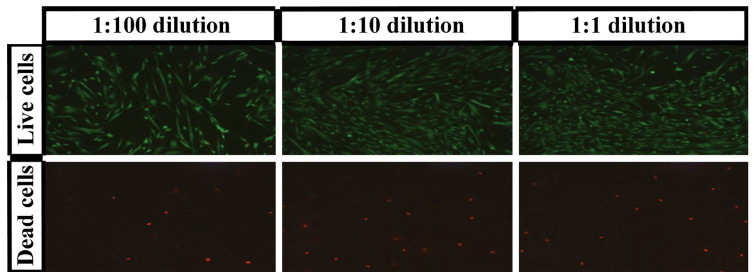
Fluorescence microscopy of live (stained with calcein AM) and dead (stained with ethidium homodimer) human fibroblast cell lines treated with hydrogel (3% CMCS and 3% PEGDA) at various dilutions for 48 h. Observations were made at room temperature (~25 °C).

**Figure 7 polymers-15-00144-f007:**
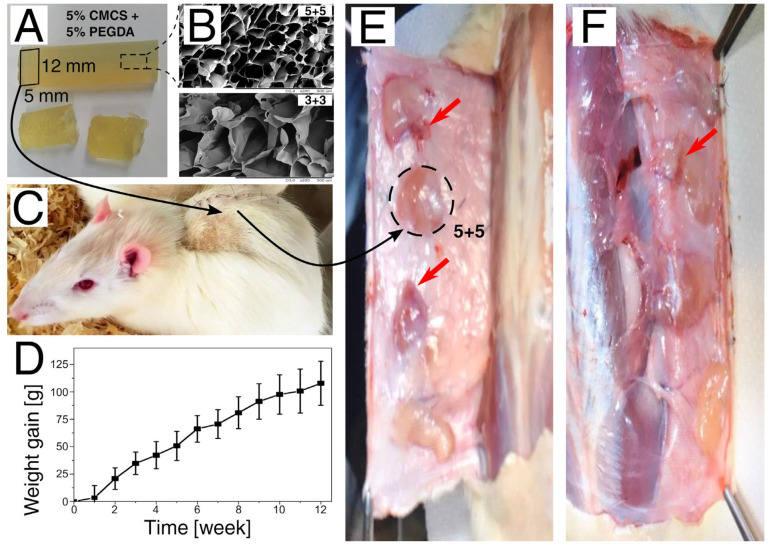
The outline of the biocompatibility test. (**A**) An exemplary hydrogel, i.e., 5% CMCS + 5% PEGDA formulation. (**B**) Images obtained by scanning electron microscopy; the upper one (5 + 5) corresponds to freeze-dried hydrogel from the first subfigure, while the bottom one highlights the microstructure differences visible in the 3% CMCS + 3% PEGDA (3 + 3) formulation. (**C**) Rat after subcutaneous implantation. (**D**) Weight gain of rats after the surgery. (**E**,**F**) Macroscopic evaluation of implantation sites (slight hyperemia is marked with red arrows).

**Table 1 polymers-15-00144-t001:** Cell viability at various concentrations of the gel extracts diluted with the pure medium in proportions of 1:1, 1:10, and 1:100.

% CMCS	% PEGDA	% Live Cells	SD (%)	% Dead Cells	SD (%)
1:1 ratio					
10%	5	95.8	7.2	8.3	1.3
	3	91.6	9.5	8.0	1.4
	2	88.1	7.2	5.8	1.4
5%	5	100.0	10.8	9.5	1.5
	3	94.7	7.4	7.8	1.8
	2	90.5	10.7	7.1	1.8
3%	5	98.2	9.8	9.2	1.6
	3	99.7	5.3	7.8	0.6
	2	90.0	5.0	5.1	1.1
1:10 ratio					
10%	5	92.8	6.1	6.7	1.5
	3	97.6	8.8	6.8	1.7
	2	99.9	8.6	6.6	1.8
5%	5	100.7	11.8	9.5	1.2
	3	95.6	9.7	7.5	2.0
	2	92.7	9.5	7.7	1.2
3%	5	100.8	7.0	7.4	1.0
	3	110.8	7.0	9.4	1.1
	2	93.8	3.4	5.6	1.7
1:100 ratio					
10%	5	92.7	5.6	6.4	1.5
	3	96.6	8.4	7.0	1.2
	2	96.9	7.2	7.2	1.1
5%	5	103.3	10.9	9.3	1.7
	3	84.6	10.6	6.5	1.3
	2	90.3	11.2	10.0	1.3
3%	5	100.0	5.3	8.9	1.0
	3	110.4	9.8	9.0	1.9
	2	99.3	7.9	6.3	1.8
Negative control		100.0	4.65	0.0	0.15
Positive control		0.0	1.21	100.0	3.26

## Data Availability

The data presented in this study are available on request from the corresponding authors.

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
