# Peer review of "Biocompatibility and Mechanical Properties of Carboxymethyl Chitosan Hydrogels"

_polymers, 2022, doi:10.3390/polym15010144_

Round 1

Reviewer 1 Report

The authors have demonstrated that chitosan-based hydrogels, consisting of varying concentrations of the carboxymethyl chitosan and crosslinking agent (polyethylene glycol diacrylate), could be potentially used as wound healing & dressing materials. The mechanical properties of the hydrogels were tuned by mainly controlling the concentrations of components or the ionizing radiation process. The biocompatibility of hydrogels was tested both in vitro and in vivo.

However, there are a few points remain unclear according to the authors' claims from the context:

Some overall issues:

I would recommend the authors use professional graphing tools such as Prism or Origin rather than using excel for research figure generation. If repetitive experiments were done in each sample, you should adequately provide the means, error bars, and p values among groups. Scientific notation should always be used in each and every figure.

In particularly:

If you are facing high dimensional data as your results in mechanical testing, you may consider proper visualization tools such as heat maps, pairs plots and etc. Figure 4-6 could be much better concluded and accessible. Figure 3 should be supplementary or not presented as an exemplary result of one group is never a major scientific find out from a paper.

The in vitro viability test followed the rule that the total percentage of live and dead cells should be 100%. The results from control groups should also be given. Line 165-166 & 391, I'm not sure if understand you correctly that water, not PBS or any buffers, was used for cell culture.

The in vivo observation end-point was unclear. The weight gain of implanted mice did not necessarily conclude that the hydrogel was biocompatible. Inflammation, immune suppression and etc. could be factors affecting the mice's weight and survival. In addition, the subcutaneous implantation of seven hydrogels in each mouse brought unmeasurable batch differences into the experiment and control groups. Also, the description of hydrogel integration should have visualized evidence to support them. In line 415, never describe anything subjectively, let alone the feelings of mice. Line 424, supplementary evidence needed. 

Author Response

Dear Reviewer 1, thank you for all your valuable comments. For our responses, please see the attachment. 

Reviewer 2 Report

The manuscript entitled “Biocompatibility and mechanical properties of carboxymethyl chitosan hydrogels for wound healing therapy” presents chitosan based hydrogels cross-linked with poly(ethylene glycol)diacrylate. The author evaluated the biocompatibility and mechanical properties of these chitosan hydrogels.  However, some issues needs to be addressed before publication. Therefore, I suggest acceptance in this journal after major revision.

Comments:

1. Although synthesis of carboxymethyl chitosan-based hydrogels via ionizing radiation has been reported, the mechanisms of gelling process with PEGDA should be explained and discussed.

2. The photographs of the hydrogels should be provided.

3. The typical porous structure of hydrogels should be evaluated.

4. For in vivo biocompatibility test, the author only provided the data of Average weight gain of rats, the histological assay of the implantation sites or the main organs of the rats should be given to prove the good in vivo biocompatibility of these hydrogels.

5. The author did not show the effect of the hydrogels on would healing, therefore the title of the manuscript should be modified, remove the words ‘for wound healing therapy’  

Author Response

Dear Reviewer 2, thank you for all your valuable comments. For our responses, please see the attachment.

Reviewer 3 Report

The paper entitled “Biocompatibility and mechanical properties of carboxymethyl chitosan hydrogels for wound healing therapy” deals with an interesting and equally popular topic. In my opinion it contains quality chosen techniques for characterization hydrogels. However, I have some prepositions in order to improve the manuscript. Firstly, it is clear that you performed mechanical tests in order to select the best samples for the further research, but it would be more effective if you did this tests at skin temperature, 32°C. Also, it would be useful if you included rheological tests in the work. The discussion should contain part related to the spreadability and compactness of the samples. In Figure legends it would be important to include at which temperature the measurements were done.

Author Response

Dear Reviewer 3, thank you for all your valuable comments. For our responses, please see the attachment.

Round 2

Reviewer 2 Report

-

Author Response

Dear Reviewer 2, we are grateful for all your previous comments. We understand that no additional work is required from Authors. Thank you so much.

Reviewer 3 Report

In my opinion the manuscript entitled „Biocompatibility and mechanical properties of carboxymethyl chitosan hydrogels for wound healing therapy“ deserves to be published in its revised form. 

Author Response

Dear Reviewer 3, we are grateful for all your previous comments. We understand that no additional work is required from Authors. Thank you so much.